# Empirical examination of the replicability of associations between brain structure and psychological variables

Shahrzad Kharabian Masouleh[1,2]*, Simon B Eickhoff[1,2], Felix Hoffstaedter[1,2], Sarah Genon[1]*, Alzheimer's Disease Neuroimaging Initiative

[1]Institute of Neuroscience and Medicine (INM-7: Brain and Behaviour), Research Centre Jülich, Jülich, Germany; [2]Institute of Systems Neuroscience, Heinrich Heine University Düsseldorf, Düsseldorf, Germany

**Abstract** Linking interindividual differences in psychological phenotype to variations in brain structure is an old dream for psychology and a crucial question for cognitive neurosciences. Yet, replicability of the previously-reported 'structural brain behavior' (SBB)-associations has been questioned, recently. Here, we conducted an empirical investigation, assessing replicability of SBB among heathy adults. For a wide range of psychological measures, the replicability of associations with gray matter volume was assessed. Our results revealed that among healthy individuals 1) finding an association between performance at standard psychological tests and brain morphology is relatively unlikely 2) significant associations, found using an exploratory approach, have overestimated effect sizes and 3) can hardly be replicated in an independent sample. After considering factors such as sample size and comparing our findings with more replicable SBB-associations in a clinical cohort and replicable associations between brain structure and non-psychological phenotype, we discuss the potential causes and consequences of these findings.
DOI: https://doi.org/10.7554/eLife.43464.001

*For correspondence:
s.kharabian@fz-juelich.de (SKM);
s.genon@fz-juelich.de (SG)

Competing interests: The authors declare that no competing interests exist.

## Introduction

The early observations of inter-individual variability in human psychological skills and traits have triggered the search for defining their correlating brain characteristics. Studies using in-vivo neuroimaging have provided compelling evidence of a relationship between human skills and traits and brain morphometry that were further influenced by individuals' years of experience, as well as level of expertise. More subtle changes were also shown following new learning/training (*Draganski et al., 2004*; *Taubert et al., 2011*), hence further demonstrating dynamic relationships between behavioral performance and brain structural features. Such observations quickly generated a conceptual basis for growing number of studies aiming to map subtle inter-individual differences in observed behavior such as personality traits (*Nostro et al., 2017*), impulsivity traits (*Matsuo et al., 2009*) or political orientation (*Kanai et al., 2011*) to normal variations in brain morphology (for review see *Genon et al., 2018*; *Kanai and Rees, 2011*). Altogether, these studies created an empirical background supporting the assumption that the morphometry of the brain in humans is related to the wide spectrum of aspects observed in human behavior. Such reports on structural brain behavior (SBB) associations may not only have important implications in psychological sciences and clinical research (*Ismaylova et al., 2018*; *Kim et al., 2015*; *Luders et al., 2013*; *Luders et al., 2012*; *McEwen et al., 2016*), but also possibly hold an important key for our understanding of brain functions (*Genon et al., 2018*) and thus concern many research fields including basic cognitive neuroscience.

**eLife digest** All human brains share the same basic structure. But no two brains are exactly alike. Brain scans can reveal differences between people in the organization and activity of individual brain regions. Studies have suggested that these differences give rise to variation in personality, intelligence and even political preferences. But recent attempts to replicate some of these findings have failed, questioning the existence of such a direct link, specifically between brain structure and human behavior. This had led some disagreements whether there is a general replication crisis in psychology, or if the replication studies themselves are flawed.

Kharabian Masouleh et al. have now used brain scans from hundreds of healthy volunteers from an already available dataset to try to resolve the issue. The volunteers had previously completed several psychological tests. These measured cognitive and behavioral aspects such as attention, memory, anxiety and personality traits. Kharabian Masouleh et al. performed more than 10,000 analyzes on their dataset to look for relationships between brain structure and psychological traits. But the results revealed very few statistically significant relationships. Moreover, the relationships that were identified proved difficult to replicate in independent samples.

By contrast, the same analyzes demonstrated robust links between brain structure and memory in patients with Alzheimer's disease. They also showed connections between brain structure and non-psychological traits, such as age. This confirms that the analysis techniques do work. So why did the new study find so few relationships between brain structure and psychological traits, when so many links have been reported previously? One possibility is publication bias. Researchers and journals may be more likely to publish positive findings than negative ones.

Another factor could be that that most studies use too few participants to be able to reliably detect relationships between brain structure and behavior, and that studies with 200 to 300 participants are still too small. Therefore, future studies should use samples with many hundreds of participants, or more. This will be possible if more groups make their data available for others to analyze. Researchers and journals must also be more willing to publish negative findings. This will help provide an accurate view of relationships between brain structure and behavior.
DOI: https://doi.org/10.7554/eLife.43464.002

Yet, along with the general replication crisis affecting psychological sciences (*Button et al., 2013*; *De Boeck and Jeon, 2018*; *Open Science Collaboration, 2015*), replicability of the previously reported SBB-associations were also questioned recently. In particular, (*Boekel et al., 2015*) in a purely confirmatory replication study, picked on few specific previously reported SBB-associations. Strikingly, for almost all the findings under scrutiny, they could not find support for the original results in their replication attempt.

In another study we demonstrated lack of robustness of the pattern of correlations between cognitive performance and measures of gray matter volume (GMV) in a-priori defined sub-regions of the dorsal premotor cortex in two samples of healthy adults (*Genon et al., 2017*). In particular we found a considerable number of SBB-associations that were counterintuitive in their directions (i.e., higher performance related to lower gray matter volume). Furthermore, subsampling revealed that for a given psychological score, negative correlations with GMV were as likely as positive correlations. Although our study did not primarily aim to address the scientific qualities of SBB, it revealed, in line with *Boekel et al. (2015)*, that a replication issue in SBB-associations could seriously be considered. However, ringing the warning bell of a replication crisis would be premature since these previous studies have approached replicability questions within very specific contexts and methods and using small sample sizes (*Muhlert and Ridgway, 2016*).

In particular, Boekel et al. and Genon et al.'s studies were performed by focusing on a-priori defined regions-of-interest (ROIs). However, several SBB studies are commonly performed in groups of dozens of individuals, using an exploratory setting employing a mass-univariate approach. Thus, the null findings of the two questioning studies could be related to the focus and averaging of GMV within specific regions-of-interest, as suggested by *Kanai (2016)* and discussed in *Genon et al. (2017)*.

In stark contrast with this argument, in whole-brain mass-univariate exploratory SBB studies, the multitude of statistical tests that is performed (as the associations are tested for each voxel, separately) likely yield many false positives. Directly addressing this limitation, several strategies for multiple comparison correction have been proposed to control the rate of false positives (*Eklund et al., 2016*). We could hence assume that the high number of multiple tests and general low power of neuroimaging studies, combined with the flexible analysis choices (*Button et al., 2013*; *Poldrack et al., 2017*; *Turner et al., 2018*) represent critical factors likely to lead to the detection of spurious and not replicable associations.

Characterization of spatial consistency of findings across neuroimaging studies is often performed with meta-analytic approaches, pooling studies investigating similar neuroimaging markers in relation to a given behavioral function or condition. However, in the case of SBB, the heterogeneity of the behavioral measures and the large proportion of apriori-ROI analyses complicate the application of a meta-analytic approach. Illustrating these limitations, previous meta-analyses have focused on specific brain regions and capitalized on a vast majority of ROI studies. For example, (*Yuan and Raz, 2014*) have focused on SBB within the frontal lobe based on a sample made of approximately 80% of ROI studies. Given these limitations of meta-analytic approaches for the SBB literature, an empirical evaluation of the replicability of the findings yielded by an exploratory approach is crucially needed to allow questioning the replicability of exploratory SBB studies.

Thus in the current study, we empirically examined replicability rates of SBB-association over a broad range of psychological scores, among heathy adults. In order to avoid the criticisms raised regarding the low sample size in Boekel et al.'s study, we used an openly available dataset of a large cohort of healthy participants and assessed replication rate of SBB-associations using both an exploratory as well as a confirmatory approach. While in the recent years multivariate methods are frequently recommended to explore the relationship between brain and behavior (*Cremers et al., 2017*; *Smith and Nichols, 2018*), SBB-association studies using these approaches remain in minority. The mass-univariate approach is still the main workhorse tool in such studies, not only due to its historical precedence and its wide integration in common neuroimaging tools, but also possibly owing to more straightforward interpretability of the detected effects (*Smith and Nichols, 2018*). The current study, therefore, focused on the assessment of replicability of SBB-associations using the latter approach.

In particular, we first identified 'significant' findings with an exploratory approach based on mass-univariate analysis, searching for associations of GMV with psychometric variables across the *whole brain*. Here a linear model was fit between inter-individual variability in the psychological score and GMV at each voxel. Inference was then made at cluster level, using a threshold-free cluster enhancement approach (*Smith and Nichols, 2009*). We then investigated the reproducibility of these findings, across resampling, by conducting a similar whole-brain voxel-wise exploratory analysis within 100 randomly generated subsamples of individuals (discovery samples). Each of these 100 discovery subsamples (of the same size) were generated by randomly selecting apriori-defined number of individuals (e.g. 70%) from the original cohort under study. In order to empirically investigate spatial consistency of significant results from these 100 exploratory analyses, an aggregate map characterizing the spatial overlap of the significant findings across all discovery samples was generated. This map denotes the frequency of finding a *significant* association between the behavioral score and gray matter volume, at each voxel, over 100 analyses and thus provides information about replicability of 'whole brain exploratory SBB-associations' for each behavioral score. Conceptually, this map gives an estimate of the spatial consistency of the results that one could expect after re-running 100 times the same SBB study across similar samples.

Additionally, for each of the 100 exploratory analyses, we assessed the replicability of SBB-associations using a confirmatory approach (i.e. ROI-based approach). For each of the 100 discovery samples, we generated a demographically-matched test pair sample from the *remaining* participants of the main cohort. Average GMV within regions showing significant SBB-association in the initial exploratory analysis, that is ROIs, are calculated among the demographically-matched independent sample and their association with the same psychological score was compared between the discovery and matched-replication sub-samples (see Materials and methods for more details).

Confirmatory replication is commonly used in the literature (*Boekel et al., 2015*; *Genon et al., 2017*; *Open Science Collaboration, 2015*), nevertheless, there is no single standard defined for evaluating the replication success. Therefore, here, we assessed the replication rate of SBB, for three

different definitions of successful replication in the confirmatory analyses: 1- Successful replication of the direction of association, only; 2- Detection of significant (p<0.05) association in the same direction as the exploratory results; While the first definition is arguably too lenient and may result in many very small correlation coefficients defined as successful replication, it is frequently used as a qualitative measure of replication and may be used to characterize the possible inconsistency of the direction of associations (that was observed in our previous study [*Genon et al., 2017*]). In addition it could be used as a complement for the possible limitation of the second definition, namely the possibility of declaring many replications that fell just short of the bright-line of p<0.05 as failed replication. 3- lastly, in line with previous studies and the reproducibility literature, we included the Bayes Factors (BF) to quantify evidence that the replication sample provided in favor of existence or absence of association in the same direction than in the discovery subsample (*Boekel et al., 2015*). In other words, when compared to standard p-value methodology, here hypothesis testing using BF enables additional quantification of the evidence in favor of the null hypothesis, that is evidence for the absence of a correlation; see Materials and methods for more details.

If the replication issue of SBB associations can be objectively evidenced, this naturally opens the questions of the accounting factors. Here, we considered proximal explanatory factors, in particular at the measurements and analysis level, but also in relation to the object level, that is, in relation to the nature itself of variations in brain structure and psychometric scores in healthy individuals. One main proximal factor that is almost systematically blamed is small sample size. In line with replication studies in other fields (e.g. *Cremers et al., 2017*; *Turner et al., 2018*), we thus here investigated the influence of sample size and replication power on the reproducibility of SBB-associations. More specifically for every phenotypic score under study we repeated both whole brain exploratory and ROI-based confirmatory replication analyses using three sample sizes (see Materials and methods for more details) to assess how sample size influences replication rate of SBB. Furthermore, for the successfully replicated effects, we also investigated existence of a positive relationship between the effect size of exploratory and confirmatory analyses.

Finally, in order to promote discussion on the underlying reality which is aimed to be captured by SBB in the framework of the psychology of individual differences, we included as benchmarks non-psychological phenotypical measures, that is age and body-mass-index (BMI), and extended our analysis to a clinical sample, where SBB-associations are expected to enjoy higher biological validity. For this purpose, a subsample of patients drawn from Alzheimer's Disease Neuroimaging Initiative (ADNI) database were selected, in which replicability of structural associations of immediate-recall score from Rey auditory verbal learning task (RAVLT) (*Schmidt, 1996*) was assessed (see Materials and methods). Due to availability of the same score within the healthy cohort, this later analysis is used as a 'conceptual' benchmark.

## Results

A total of 10800 exploratory whole brain SBB associations (each with 1000 permutations) were tested to empirically identify the replicability of the associations of 36 psychological scores with GMV over 100 splits in independent matched subsamples, at three pre-defined sample sizes, within the *healthy* cohort; see *Supplementary file 1*, for total number of participants with available score for each of the psychological scores.

Altogether, in contrast to GMV-associations with age and BMI, significant SBB-associations were highly unlikely. For the majority of the tested psychological variables no significant association with GMV were found in more than 90% of the whole brain analyses.

### SBB-associations among the healthy population

#### Replicability of 'whole brain exploratory SBB-associations'

Age and BMI structural associations: Voxel-wise associations of age and BMI with GMV, as suggested by previous studies (*Fjell et al., 2014*; *Kharabian Masouleh et al., 2016*; *Salat et al., 2004*; *Willette and Kapogiannis, 2015*), were widespread and strong.

Despite using more stringent thresholds, compared to the threshold used for the psychological scores (see Materials and methods), for almost all subsamples, we found highly consistent widespread negative associations of GMV with age. See *Figure 1A* for aggregate maps of spatial overlap

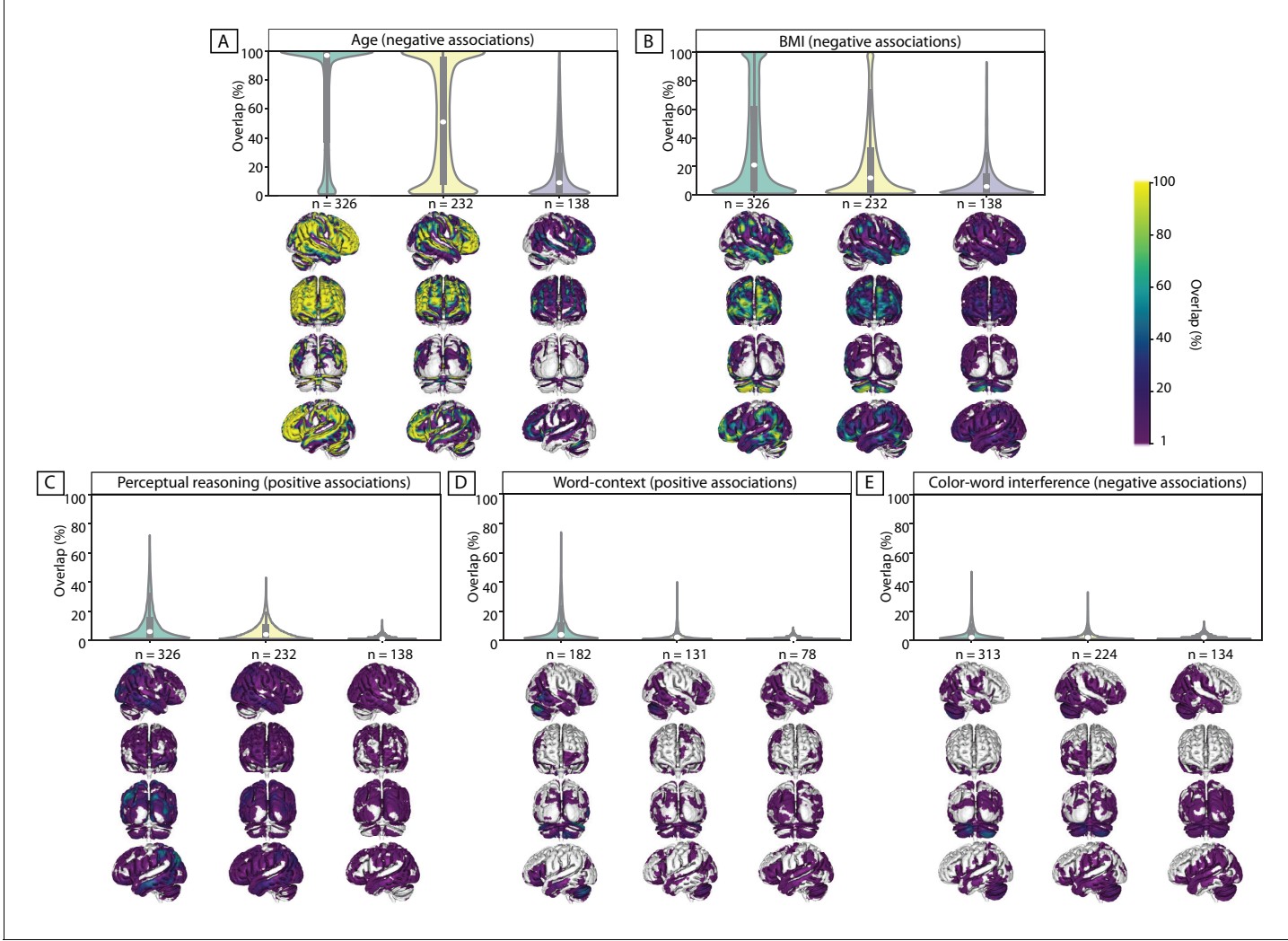

**Figure 1.** Replicability of exploratory results within healthy cohort. Frequency of spatial overlap (density plots and aggregate maps) of significant findings from exploratory analysis over 100 random subsamples are depicted for few behavioral score. For each score, columns show the results of three different discovery sample sizes (i.e. when discovery cohorts are generated from 70%, 50% or 30% of the main sample, from left to right respectively (x-axis)). The density plots show the distribution of values within their corresponding aggregate map. The y-axis depicts the frequency of spatial overlap (in %) and the density plots show the distribution of values within their corresponding aggregate maps. In addition to age and BMI (**A**, **B**), which are used as benchmarks, the top three behavioral scores with the highest frequency of overlapping findings are depicted (**C–E**). Within each density plot, the box-plot shows the quartiles and extent of the distribution and the white dot depicts the median of percentage of overlap. On the spatial maps, lighter colors denote higher number of samples with a significant association at the respective voxel. BMI: body mass index; CWI: color-word interference; n = number of participants within the discovery samples.

DOI: https://doi.org/10.7554/eLife.43464.003

of exploratory findings and density plots, summarizing distribution of 'frequency of significant findings' within each map.

When decreasing the sample size of the discovery cohort, the spatial overlap of significant findings over 100 splits decreased. More specifically, for the discovery sample of 326 subjects, more than half of the significant voxels were consistently found as being significant in beyond 90% of the whole-brain exploratory analyses (i.e. high level of spatial consistency of significant findings). As the size of the subsamples decreased, the shape of the distribution also changed, and the median of the density plots fell around 50% and even 10% for samples consisting of 232 and 138 individuals, respectively.

Similar results, though with much lower percent of consistently overlapping voxels, were seen for negative associations of BMI with GMV. The density plots and the spatial maps of **Figure 1B** show

that for the larger samples (consisting of 326 and 232 subjects) few voxels were consistently found in 'all' (100%) subsamples as having significant negative association with BMI. For the smaller samples (with 138 participants) the maximum replicable association was found in 93% of the splits and 4 out of 100 exploratory analyses did not result in any significant clusters (*Table 1*). Additionally, as *Figure 2B* shows, the majority of significant voxels had a replicability bellow 50%.

These results highlight the influence of sample size on the replicability (frequency of overlap) of whole-brain significant associations, even for age and BMI, for which we expected more stable associations with morphological properties of the brain.

Structural associations of the psychological scores: In contrast, for most of the psychological scores, only few of the 100 discovery subsamples yielded significant clusters. *Table 1* and *Supplementary file 2* show the number of splits for which the exploratory whole-brain SBB-analysis resulted in *at least one* significant positively or negatively associated cluster for each score. These results reveal that finding significant SBB-associations using the exploratory approach in healthy individuals is highly *unlikely* for most of the psychological variables. Furthermore, the significant findings were spatially very diverse, that is, spatially overlapping findings were very rare.

We here retained for further analyses the three psychological scores for which the discovery samples most frequently resulted in at least one significantly associated cluster. These three scores were the Perceptual reasoning score of WASI (*Wechsler, 1999*), the number of correct responses in word-context test and the interference time in the color-word interference task. For example, for the discovery samples of 326 adults, in 83 out of 100 randomly generated discovery samples, at least one cluster (not necessarily overlapping) showed a significant positive association between perceptual reasoning and GMV (*Table 1*)). Of note, these more frequently found associations were in the direction linking better task performance with higher GMV.

Yet again, in line with our observations for BMI associations, the probability of finding at least one significant cluster tend to decrease in smaller discovery samples (see *Table 1*). Likewise, as the discovery sample size decreased, the maximum rate of spatial overlap, as denoted by the height of the density plots, decreased (see *Figure 1C–F*). The width of these plots show that the majority (>50%) of the significant voxels spatially overlapped only in less than 10% of the discovery samples. In the same line, the variability depicted by the spatial maps highlight that many voxels are found as significant only in one out of 100 analyses.

**Table 1.** Summary of exploratory findings.

For each discovery sample size, the number of clusters in which gray matter volume is positively or negatively associated with the tested phenotypic or psychological score is reported. The number of splits (out of 100) in which the clusters were detected are noted in parentheses (i.e. % of splits with at least one significant cluster [in the respective direction]).

| Healthy cohort | n_discovery = 70% n_total | | n_discovery = 50% n_total | | n_discovery = 30% n_total | |
|---|---|---|---|---|---|---|
| | # positively associated clusters (split%) | # negatively associated clusters (split%) | # positively associated clusters (split%) | # negatively associated clusters (split%) | # positively associated clusters (split%) | # negatively associated clusters (split%) |
| Age (years) n-total = 466 | 77 (54%) | 154 (100%) | 5 (4%) | 522 (100%) | 1 (1%) | 1781 (100%) |
| BMI (kg/m$^2$) n-total = 466 | 0 | 1741 (100%) | 0 | 2276 (100%) | 0 | 1937 (96%) |
| Perceptual IQ (sum of t-scores) n-total = 466 | 499 (83%) | 0 | 256 (58%) | 0 | 145 (33%) | 0 |
| Word-context (# of consecutively correct) n-total = 262 | 337 (80%) | 0 | 159 (47%) | 0 | 80 (21%) | 0 |
| CWI (interference) (sec) n-total = 449 | 0 | 163 (53%) | 1 (1%) | 122 (39%) | 6 (1%) | 60 (26%) |
| **Clinical cohort** | - | | n_discovery = 50% n_total | | - | |
| RAVLT (# total immediate recall) | - | - | 309 (84%) | 0 | - | - |

Abbreviations: BMI: body mass index; IQ: intelligence quotient, CWI: color-word interference task; RAVLT: Rey auditory verbal learning task;
DOI: https://doi.org/10.7554/eLife.43464.006

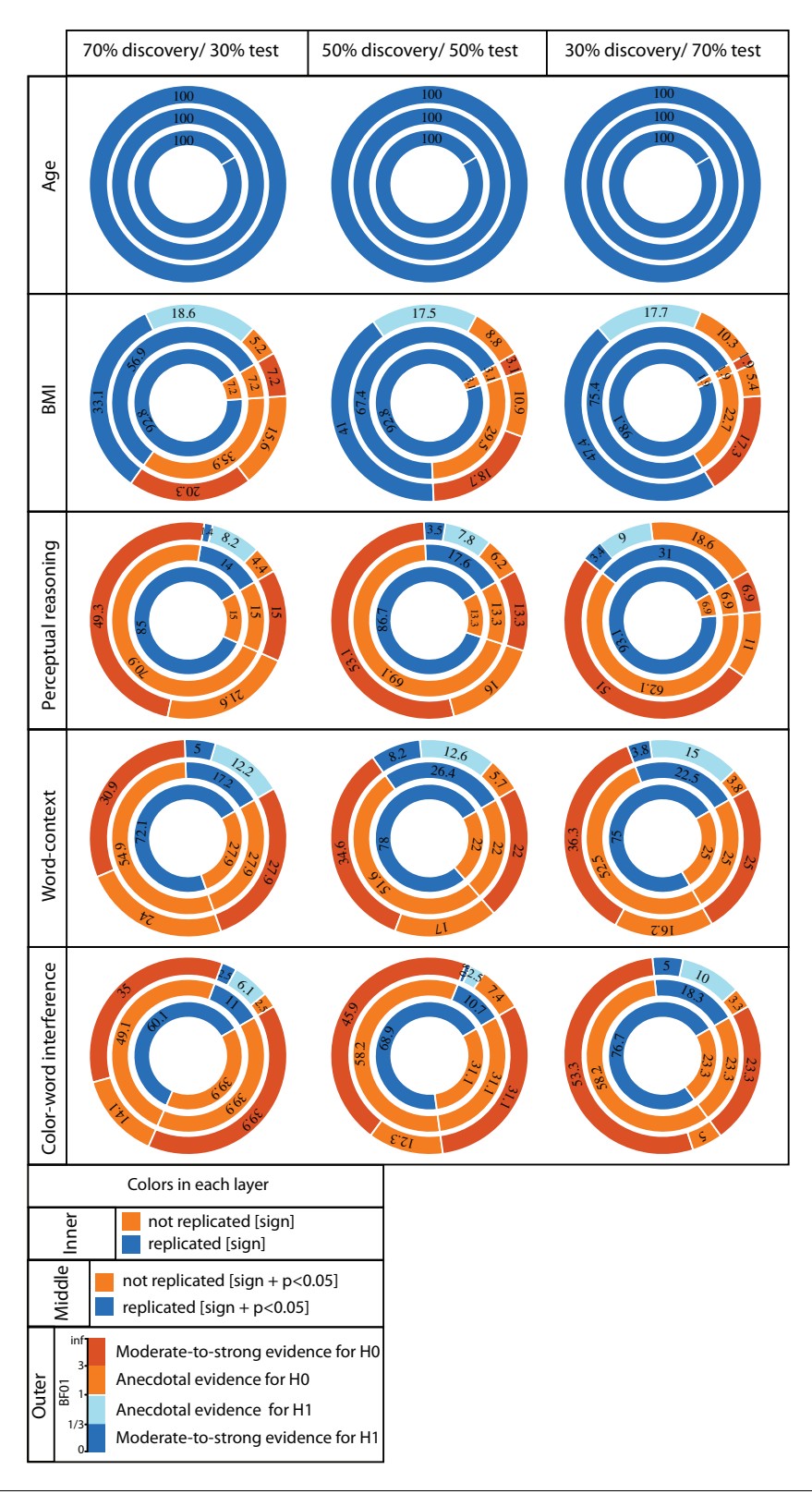

**Figure 2.** ROI-based confirmatory replication results within healthy cohort. Donut plots summerising ROI-based replication rates (% of ROI) using three different critera for three different sample sizes among heathy participants. The most inner layers depict replication using 'sign' only (blue: replicated, orange: not replciated). The middle layers define replication based on similar 'sign' as well as 'statistical significance' (i.e. p<0.05) (blue: replicated,

*Figure 2 continued on next page*

*Figure 2 continued*
orange: not replciate). The most outer layers define replication using 'bayes factor' (blue: "moderate-to-string evidece for H1, light blue: anecdotal evidence for H1; light orange: anecdotal evidence for H0, orange: "moderate-to-string evidece for H0).
DOI: https://doi.org/10.7554/eLife.43464.004
The following figure supplement is available for figure 2:

**Figure supplement 1.** ROI-based confirmatory replication results for five personality subscores within healthy cohort.
DOI: https://doi.org/10.7554/eLife.43464.005

These results highlight that finding a significant association between normal variations on behavioral scores and voxel-wise measures of GMV among healthy individuals is highly unlikely, for most of the tested domains. Furthermore, they underscore the extent of spatial inconsistency and the *poor replicability* of the significant SBB-associations from *exploratory analyses*.

## Confirmatory ROI-based SBB-replicability

Age and BMI effects: Irrespective of the size of the test subsamples and definition used to identify 'successful' replication (see Materials and methods), for all ROIs negative age-GMV associations were 'successfully' replicated in the matched test samples. Unlike the perfect replication of age-associations, replication rate of BMI effects depended highly on the test sample size and the criteria used to characterize 'successful' replication. Over all three tested sample sizes, in more than 90% of the a-priori defined ROIs, BMI associations were found to be in the same 'direction' in the discovery and test samples (i.e. replicated based on 'sign' criteria). The examination of replicated findings based on 'statistical significance' revealed replicated effects in more than 57% of ROIs. This rate of ROI-based replicability increased from ~57% to 75%, as the test sample size increased from 140 to 328 individuals (see *Figure 2*). Furthermore, as the dark blue segments in the outer layers of *Figure 2* indicates, Bayesian hypothesis testing revealed moderate-to-strong evidence for H1 in more than 30% of the ROIs.

Psychological variables: *Figure 2* also illustrates the replicability rates of structural associations of the top three psychological measures from the whole brain analyses (the perceptual reasoning score of WASI, the number of correct responses in word-context test and the interference time in the color-word interference task).

Despite the structural associations of perceptual reasoning score being in the same direction (positive SBB-association), for the majority of the ROIs (>85%), less than 31% of all ROIs showed replicated effects based on 'statistical significance' criterion. Finally, less than 4% of the ROIs were identified as 'successfully replicated' based on the Bayes factors. (*Figure 2*).

For the three tested samples sizes, associations of the word-context task were in the same direction (positive SBB-association) in the discovery and test pairs in ~75% of ROIs. Nevertheless, again, the rate of statistically 'significantly'-replicated ROIs ranged between 17% to 26%. Furthermore, even less than 8% of all ROIs showed replicated effects based on the Bayes factors (moderate-to-strong evidence for H1) (*Figure 2*).

Finally, negative correlations between interference time of the color-word interference task and average GMV were depicted in ~70% of the ROIs, but significant-replication was found in only 11% to 17% of all ROIs, for the three test sample sizes. Along the same line, replication based on the Bayes factors was below 5% (*Figure 2E*).

In general, these results show the span of replicability of structural associations from highly replicable age-effects to very poorly replicable psychological associations. They also highlight the influence of the sample size, as well as the criteria that is used to define successful replication on the rate of replicability of SBB-effects in independent samples.

## Effect size in the discovery sample and its link with effect size of the test sample and actual replication

*Figure 3* plots discovery versus replication effect size (i.e. correlation coefficient) for each ROI and for three test sample sizes. Focusing on by-'sign' replicated ROIs (blue), for the three psychological scores (perceptual reasoning, word-context and CWI) revealed that the discovery samples resulted in overall larger effects (magnitude) compared to the test samples. Indeed, the marginal

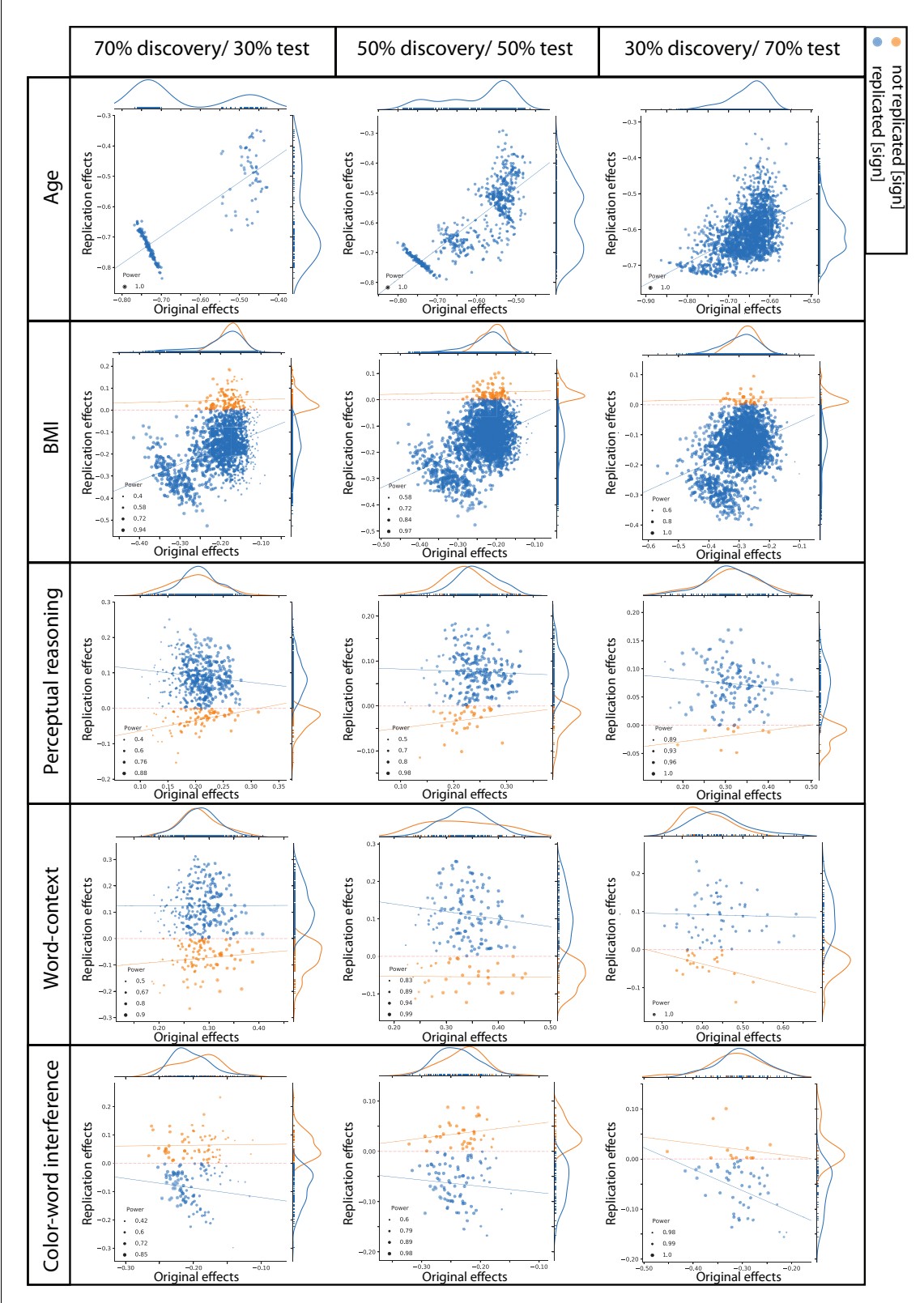

**Figure 3.** Discovery versus replication effects sizes: Scatter plots of correlation coefficients in the discovery versus replication sample for all ROIs from 100 splits within healthy cohort; each point denotes one ROI, which is color-coded based on its replication status (by-'sign'). The size of each point is proportional to its estimated statistical power of replication. Regresion lines are drawn for the replicated and unreplicated ROIs, separately.

DOI: https://doi.org/10.7554/eLife.43464.007

distributions are centered around smaller correlation coefficients in the y-dimension (test sample) compared to the x-axis (discovery samples). Furthermore, for these by-'sign' replicated ROIs, there was no positive relationship between the effect sizes of the behavioral associations in the discovery and test samples (blue lines in each subplot).

For BMI and age, however, the effect sizes of the discovery and test pairs were generally positively correlated, suggesting that the ROIs with greater negative structural association with BMI (or age) in the discovery sample, also tended to show stronger negative associations within the matched test sample.

To investigate if the replication power, estimated using the correlation coefficient within the discovery samples, was linked to a higher probability of *actual* replication in the test samples, the ROIs were grouped into replicated and not-replicated, based on the 'statistical significance' criterion. While the estimations of statistical power were generally higher among the replicated compared to not-replicated ROIs for BMI associations (p-value of the Mann-Whitney U tests $<10^{-5}$), for structural associations of the psychological scores, this was not the case. Strikingly, for the structural associations of perceptual reasoning, over all sample sizes, the significantly replicated ROIs tended to have lower estimated power compared to the ROIs that actually were not-replicated (p-value of the Mann-Whitney U tests $<10^{-5}$). These unexpected findings highlight the unreliable aspect of effect size estimations of SBB-associations within the discovery samples among healthy individuals. They also demonstrate that these inflated effect sizes result in flawed and thus uninformative estimated statistical power.

## Structural associations of total immediate recall score in ADNI cohort

### Replicability of 'whole brain exploratory associations'

Within the sample of patients from ADNI-cohort, 84 out of the 100 whole-brain exploratory analyses resulted in *at least one* significant cluster showing a positive association between the immediate-recall score and GMV. In the healthy population, however, the same score resulted in a significant cluster in only less than 10% of exploratory analyses, for any of the three discovery sample sizes (*Supplementary file 2* and *Figure 4—figure supplement 1*).

As could be seen in the spatial maps of *Figure 4*, significant associations in the ADNI cohort were found across several brain regions including the bilateral lateral and medial temporal lobe, the lateral occipital cortex, the precuneus, the superior parietal lobule, the orbitofrontal cortex and the thalamus. Although most of the significant voxels were found by less than 10% of the splits, some voxels in the bilateral hippocampus were found to be significantly associated with the recall score in more than 70% of the subsamples (peak of spatial overlap; see *Figure 4A,B*).

### Confirmatory ROI-based SBB-replicability

Figure 4D shows the rates of 'successful replication' of associations between the immediate-recall score and GMV within each ROI in the independent, matched-samples. As the most inner layer shows, in more than 94% of ROIs, GMV correlated positively with the recall score in the test subsamples, corroborating the 'sign' of the association in the paired-discovery samples. These correlations were significant in 72% of all ROIs. Furthermore, in more than 50% of all ROIs the correlations in the test sample supported, at least moderately, the link between higher GMV and higher recall score (using the Bayes factors).

### Association between discovery and replication effect size

The marginal histograms in *Figure 4C* suggest that overall the correlations in the discovery samples are slightly stronger than the correlations in the paired replication samples. When looking at the ROIs that were successfully replicated (by-sign), there was a positive association between the discovery and replication effect size (spearman's rho = 0.38, p-value$<10^{-11}$).

Finally, the median replication power was higher among 'significantly replicated' ROIs, compared to not replicated (defined using 'statistical significance criterion') ROIs (p-value of the mann-whiteney U test $<10^{-3}$). These results showed the superior, yet not perfect, replicability of SBB-associations within the clinical population (see *Figure 4—figure supplement 1* for structural associations of immediate recall within healthy cohort). The observed somewhat robustness of the findings in ADNI suggest that, when the population under study shows clear variations in both brain structural

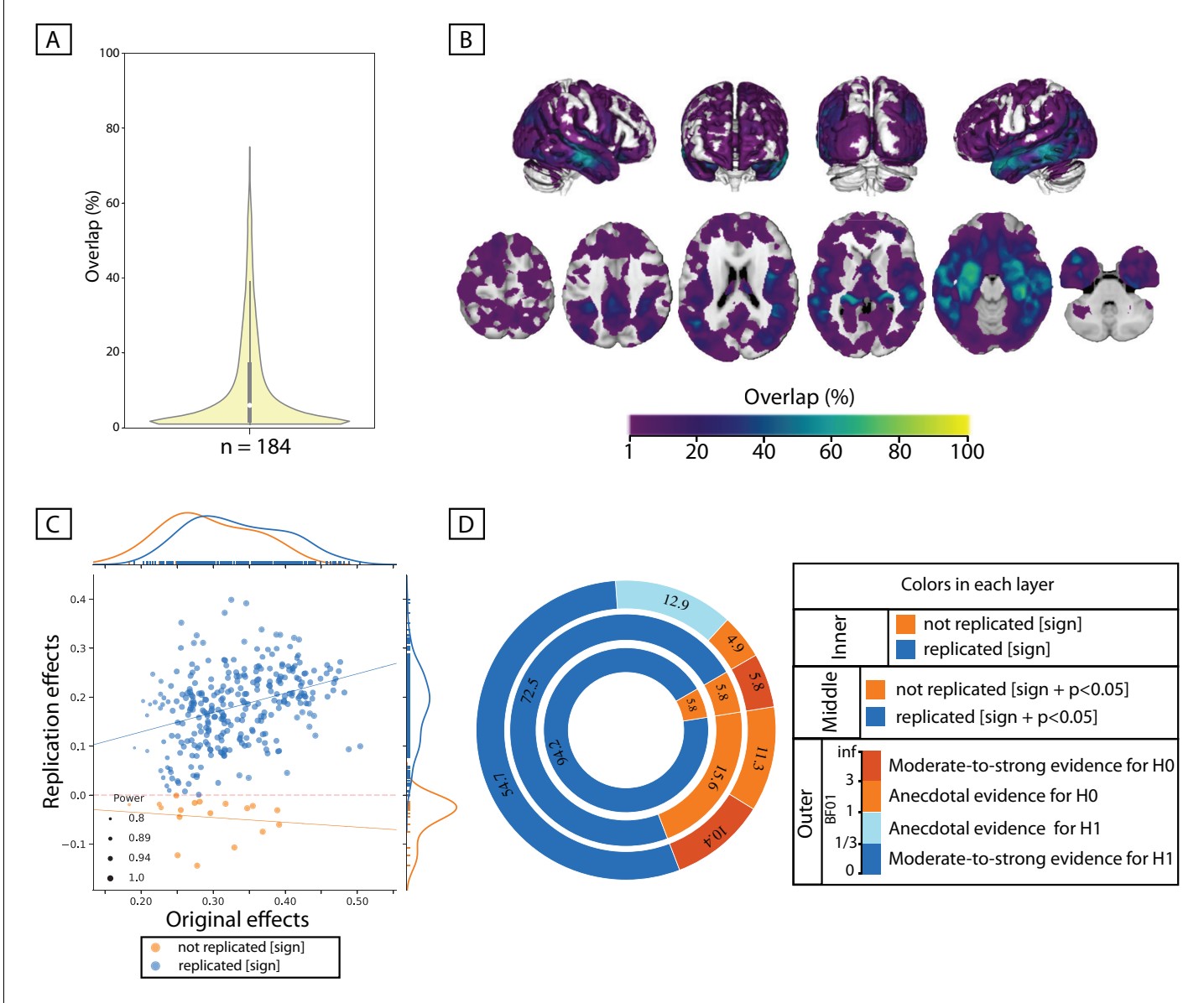

**Figure 4.** Replicability of positive association between immediate-recall and GMV within ADNI cohort. (**A, B**) Replicability of exploratory results: Frequency of spatial overlaps (density plot and aggregate maps) over 100 random subsamples. Within the density plot, the box-plot shows the quartiles and extent of the distribution and the white dot depicts the median of percentage of overlap. (**C, D**) ROI-based confirmatory replication results: C: Original versus replication effects sizes (correlation coefficient) for all ROIs from 100 splits; points are color-coded based on their replciation status (by-'sign') and size of each point is proportional to the estimated statistical power of replication. Regresion lines are drawn for the replicated and unreplicated ROIs, separately. D: Donut plots summerising ROI-based replicability rates using three different critera. The most inner layer depicts replicability using 'sign' only (blue: replicated, orange: not replciated). The middle layer, defines replication based on similar 'sign' as well as 'statistical significance' (i.e. p<0.05) (blue: replicated, orange: not replciate). The most outer layer reflects replicability using bayes factor' (blue: 'moderate-to-string evidece for H1, light blue: anecdotal evidence for H1; light orange: anecdotal evidence for H0, orange: 'moderate-to-string evidece for H0); Discovery and replication samples have equal size (n = 184) and are matched for age, sex and site.
DOI: https://doi.org/10.7554/eLife.43464.008

The following figure supplement is available for figure 4:

**Figure supplement 1.** Summary of replication of positive associations between immediate-recall and GMV within healthy cohort.
DOI: https://doi.org/10.7554/eLife.43464.009

markers and psychological measurements, such as the patient group in ADNI cohort, the associations between brain structure and psychological performance could be relatively reliably characterized. Nevertheless, again, the occurrence of not-replicated results highlight the importance of confirmatory analyses for a robust characterization of brain-behavior associations.

## Discussion

Our empirical investigation of the replicability of SBB in healthy adults showed that significant associations between psychological phenotype and GMV are not frequent when probing a range of psychometric variables with an exploratory approach. Where significant associations were found, these associations showed a poor replicability.

In the following, we first discussed implications of the very low rate of significant findings revealed by the exploratory approach. We then discussed the possible causes of the observed spatial variability of SBB-associations. Those pattern of findings are then compared with the pattern observed in the clinical cohort. Finally, in line with the replication literature in psychological sciences and neurosciences (*Button et al., 2013*; *Poldrack et al., 2017*; *Turner et al., 2018*), we devoted our last section to sample size and power issues in SBB studies in healthy adults and proposed some recommendations.

### Infrequent significant SBB associations in healthy individuals: Importance of reporting null findings

According to the scientific literature, associations between psychological phenotype (cognitive performance and psychological trait) and local brain structure are not uncommon (*Kanai and Rees, 2011*). However, in our exploratory analyses, when looking at a range of psychological variables, significant associations with GMV were very rare. It is worth noting that here by having a-priori fixed analysis design and inference routines, we aimed to avoid 'fishing' for significant findings (*Gelman and Loken, 2014*). Flexible designs and flexible analyses routines (*Simmons et al., 2011*) as well as p-hacking (*John et al., 2012*) are considered as inappropriate but frequent research practices (*Poldrack et al., 2017*). Based on our findings of infrequent significant SBB-associations, we could assume that flexible analyses routines, p-hacking and most importantly *publication bias* (*Dwan et al., 2013*) have contributed to the high number of significant SBB-reports in the literature.

When considering potential impacts of biased SBB-reports on our confidence of psychological measures, as well as our conception and apprehension of brain-behavior relationships and psychological interindividual differences, we would strongly argue for null findings reports. Such reports would contribute to a more accurate and balanced apprehension of associations between differences in psychological phenotype and brain morphometric features, but it would also help to progressively disentangle factors that mediate or modulate the relationship between brain structure and behavioral outcomes.

### Poor spatial overlap of SBB across resampling: possible causes and recommendations

In addition to the low likelihood of finding 'any' significant SBB-association using the exploratory approach, clusters that do survive the significance thresholding did not often overlap in different subsamples. Furthermore, the probability of spatial overlap further dropped as the number of participants in the subsamples decreased (*Figure 1*). Putting this finding in light of the literature brings two main hypotheses.

First, from the conceptual level, we could hypothesize that the pattern of correlation between a psychological measure is by nature spatially diffuse at the brain level. Psychological measures aim to conceptually articulate *behavioral functions and processes*, thus, in most cases, they have not been developed to identify specific localized *brain functions*. Following this philosophical segregation between psychological sciences and neurosciences, it is now widely acknowledged that there is no one-to-one mapping between behavioral functions and brain regions (*Anderson, 2016*; *Genon et al., 2018*; *Pessoa, 2014*). Instead, mapping a psychological concept to brain features usually result in a diffuse brain spatial pattern with small effect sizes (*Bressler, 1995*; *Poldrack, 2010*; *Tononi et al., 1998*). From this axiom, we can expect that several studies conducted in small samples (specifically after rigorous corrections for multiple comparisons) are likely to each capture a

partial and minor aspect of the whole true association pattern, resulting in a poor replication rate for each individual study (i.e. high type II error).

Alternatively, a more parsimonious hypothesis is a methodological one questioning the truth or validity of the found significant associations hence considering them as spurious (i.e. type I error). Psychological and MRI measurements are both relatively indirect estimations of respectively, behavioral features and brain structural features and thus are susceptible to noise. Correlations in small samples in the presence of noise for both type of variables is likely to produce spurious significant results (*Loken and Gelman, 2017*) by fitting a correlation or regression between random noise in both variables.

Thus, the pattern of poor spatial consistency of SBB findings could result either from factors at the object of study level, that is the relationship between brain and behavior, or, from factors at the measurement and analysis level. While the latter hypothesis is more parsimonious, one argument for the former hypothesis comes from the relatively substantial replications by-sign observed in our confirmatory analyses, of three top behavioral scores (see *Figure 2*). If the significant SBB findings would be purely driven by noise in the data, we would expect them to show purely random signs across resampling, which was not the case (but also see *Figure 2—figure supplement 1* for example of scores with lower replicability and higher inconsistent associations across resampling). Therefore, it is actually likely that both hypotheses hold true and that the spatial variability of significant SBB findings result from both, factors at the analyses levels and factors at the object level, potentially interacting together.

It is worth noting that similar complexity and uncertainty have been described for task-based functional associations studies (*Cremers et al., 2017*; *Turner et al., 2018*). In particular, *Cremers et al. (2017)* using simulated and empirical data demonstrated that task-based functional activations have a generally weak and diffuse pattern. Therefore, these authors concluded that most whole-brain analyses in small samples, specifically when combined with stringent correction for multiple comparison, to control the false positive rates, would most likely frequently overlook global meaningful effects and depict results with poor replicability (type II error). Relatedly, in the present study, higher spatial extent and lower consistency of significant findings in smaller samples in *Figure 1*, also suggest higher number of spurious associations (type I error) in smaller samples (due to winners curse [*Button et al., 2013*; *Forstmeier and Schielzeth, 2011*]) than in larger samples.

These factors, added to the complexity of human behavior, renders the objective of capturing covariations with psychometric variables in brain structure *locally* particularly challenging. For that reason, in exploratory studies whose aim is to identify brain structural features correlating with a specific (set of) psychological variable(s), a multivariate approach could be advised (*Habeck et al., 2010*; *McIntosh and Mišić, 2013*). As mentioned earlier, like all methods, multivariate analyses have their own limitations: in particular, the ensuing difficulty of interpretability of the revealed pattern. While some authors argue either for one or the other approach, the use of these approaches are far from being mutually exclusive (*Moeller and Habeck, 2006*). Combining both approaches in small datasets indeed revealed that the results of the univariate approach reflect the 'tip of the iceberg' of the behavior's brain correlates, whose spatial extent are more comprehensively captured with the multivariate analysis, but interpretability is facilitated by the use of univariate analyses; for example (*Genon et al., 2016*; *Genon et al., 2014*).

Thus, to partially address the previously described concerns of small and spatially diffuse effects at the brain level in exploratory whole-brain-behavior study, we here recommend for the future studies to combine a univariate and a multivariate approach. Although it does not provide any protection against the influence of noise that may affect both approaches, this solution may help to reduce the false negatives.

## Confirmatory replication of exploratory SBB findings: importance of out of sample replication

ROI-based analysis further highlighted that significant associations, which have been discovered when starting with a psychological measure and searching within the whole brain for a significant association (i.e. 'evidenced in an exploratory study'), show poor replicability (using significance and Bayes factor criteria, but also using a similar sign criterion for most psychometric scores; For example, see *Figure 2—figure supplement 1* and *Figure 4—figure supplement 1*) in a confirmatory ROI-based study (in line with what was previously shown by *Boekel et al., 2015*). These findings

thus call for a general acknowledgment of the uncertainty and fragility of exploratory findings and the need for *out of sample* confirmatory replications to provide evidence about the robustness of the reported effects (*Ioannidis, 2018*; *Tukey, 1980*).

## Further factors influencing replicability of SBB-findings: power of replication and object of study

Another clear finding of our study is the overestimation of the effect size in the exploratory approach (*Kriegeskorte et al., 2010*), specifically in smaller samples (see marginal distributions of the x- and y-axis in *Figure 3*). For the majority of the psychological scores, in the ROI-based approach, we failed to find a clear association between effect size in the discovery and replication samples. Instead, we observed a rather high estimated statistical power for replication (due to an inflated effect size estimation [*Ioannidis, 2008*]), despite very low actual rate of replicated effects in the independent samples. These findings are particularly important when considering the current research context, in which power analyses are encouraged to justify the allocation of financial and human investment in specific future researches. Prospective studies with power analyses are frequently proposed, where power is computed based on the findings from previous exploratory analyses in a small sample (*Albers and Lakens, 2018*). An inflated effect size estimation from the exploratory study results in an unreliable high power, which in turn lead to confidence in prospective studies to find relevant findings and hence in the allocation and possibly waste of (frequently public) resources (*Albers and Lakens, 2018*; *Poldrack et al., 2017*). Nevertheless, this provocative conclusion does not imply that SBB studies should be banished to hell. Our conclusion here mainly concerns the study of association between variations at *standard psychological measures* and variations in *measures of gray matter* in 'small' samples of *healthy individuals*. Our results further show that different types of SBB exploratory studies should not be epistemologically all put in the same pot.

In support for this argument, in ADNI sample, despite the additional confounding effect of different scanners and/or scanning parameters due to the multi-site nature of the cohort, associations between immediate-recall score and GMV were relatively stable. Compared to associations of the same measure of verbal learning performance within the healthy population (see supplementary Figure 1), these results highlight the superior reliability of SBB-associations that are defined in a clinical context. These findings have important conceptual implications. From an epistemological and conceptual point of view, our comparative investigation suggests that the object of study matters in the replicability of SBB. Searching for correlation between variations in cognitive performance and GMV in healthy adults, on one hand, and in neurodegenerative patients, on the other hand, appear as two different objects of study, with different replicability rates. While several SBB results in healthy population are likely to be spurious (see *Supplementary file 2*), it seems that SBB in clinical population are more likely to capture true relationships.

Thus, maybe the conceptual objective itself should be questioned: should we expect the association between normal psychological phenotype, in particular cognitive performance, in healthy population to be substantially driven by local brain macrostructure morphology? Brain structure can certainly not be questioned as the primary substrates of behavior and more than a century of lesion studies recalls this primary principle to our attention (*Broca, 1865*; *Scoville and Milner, 1957*), but this does not imply that 'normal' variations at standard psychological tests can be related to variations in markers of local brain macrostructure. Our results suggest that reliable answer to this important question requires substantially big samples (bigger than those used here) and independent replications.

## Further recommendation: Large sample sizes are important both for exploratory as well as replication analyses

The sample size and related power issues hold a central position in the current discussions of the replication crisis in behavioral sciences, as well as in neuroimaging studies (*Button et al., 2013*; *Ioannidis, 2005*; *Lilienfeld, 2017*; *Munafò et al., 2017*; *Open Science Collaboration, 2015*). Higher power is defined as increased probability of finding effects that are genuinely true. Furthermore, high power experiments have higher positive predictive values (PPV) of the claimed effects (i.e. probability that the claimed effect reflects a true effect). They also result in less exaggerated effects sizes when a true effect is discovered (*Button et al., 2013*). As such, in the discovery sample, by

increasing the sample size, the correlation coefficients get closer to their real value and their PPV increases. However, in the current study, as the number of participants in the main sample is limited, the size of the discovery and their matched replication samples are dependent on each other. Therefore, for each behavioral measure, larger discovery samples have smaller replication counterparts. These smaller replication samples have in turn lower power to find the true effects and have lower PPV. However, in splits with larger replication samples, as the discovery sample gets smaller, apart from the lower PPV, the estimated correlation coefficients are possibly more exaggerated (e.g. due to winner's curse) (*Cremers et al., 2017*) and thus the power of the replication would be over-estimated. This is a limitation which complicates the interpretation of the relationship between the calculated replication power and the actual rate of replicability of associations in the present study. We hoped that the use of a large cohort of healthy individuals as our main cohort would result in large enough discovery and test cohorts and thus minimize the impact of above-mentioned limitation. However, large discrepancies between the rate of 'significant' within-split replicability and the a-priori estimated replication power, as we observed in the ROI-based confirmatory analyses, confirms an exaggerated power estimation in most of our analyses and thus highlights the insufficiency of the size of the discovery and replication samples.

Thus overall, these findings suggest that samples consisting of ~200–300 participants have in reality still low power to identify reliable SBB-associations among healthy participants. However, the sample size of SBB studies is usually substantially smaller. *Figure 5* depicts the distribution of sample sizes (log-scale) of published studies examining GMV in human participants with the standard voxel-based morphometry approach across previous years (BrainMap data [*Vanasse et al., 2018*]). SBB studies in healthy adults also fall under this general trend. Based on our current work, we would argue that the probability of finding spurious or inconclusive results and exaggerated effect size estimations in these studies is thus quite high (*Albers and Lakens, 2018*; *Schönbrodt and Perugini, 2013*; *Yarkoni, 2009*).

In addition, to underscore the importance of the sample size, our analyses and results further show that the size of the replication sample also matters when examining the replicability of a previous SBB findings. This is an obvious factor that has been frequently neglected in the discussions about replication crisis. Yet, while many replication studies straightforwardly blame the sample size of the original studies, it is important to keep in mind that a replication failure might also come from a too small sample size of the replication study (*Muhlert and Ridgway, 2016*).

## Limitations

When interpreting our results, it should be noted that, in order to keep large sample sizes for the exploratory replication analyses, the discovery subsamples were not necessarily designed to be independent from each other. Considering this limitation, the poor spatial consistency of the whole brain exploratory associations that we observed for almost all the behavioral scores is hence even more alarming. As discussed earlier, another indirect limitation of the limited size of the selected cohort is the dependence between the size of the discovery and their matched replication sub-samples. This limitation prevents us to state strong conclusions about the relationship between the calculated replication power and the actual rate of replicability. Overall, these acknowledged limitations raise the need for even larger sample sizes for such investigations. Recent advancements through data collection from much larger number of participants, such as UK-biobank (*Miller et al., 2016*) are promising opportunities for overcoming these limitations in future replication studies.

Moreover, the generalizability of our results are partly limited to our methodological choices such as the computation of volumetric markers of brain structure (as opposed to surface-based markers), the size of the smoothing kernel, and the use of a priori-defined ROIs in the replication sample. Future studies should therefore investigate to which extend our replicability rates are reproduced with different data preprocessing pipelines and analyses approaches.

## Summary and conclusions

Overall, our work and review of the recent and concomitant replication literature in related fields demonstrate that several improvements could be recommended to get more accurate insight on the relationship between psychological phenotype and brain structure and to progressively answer open questions. Importantly, our recommendations and suggestions concern different levels of SBB

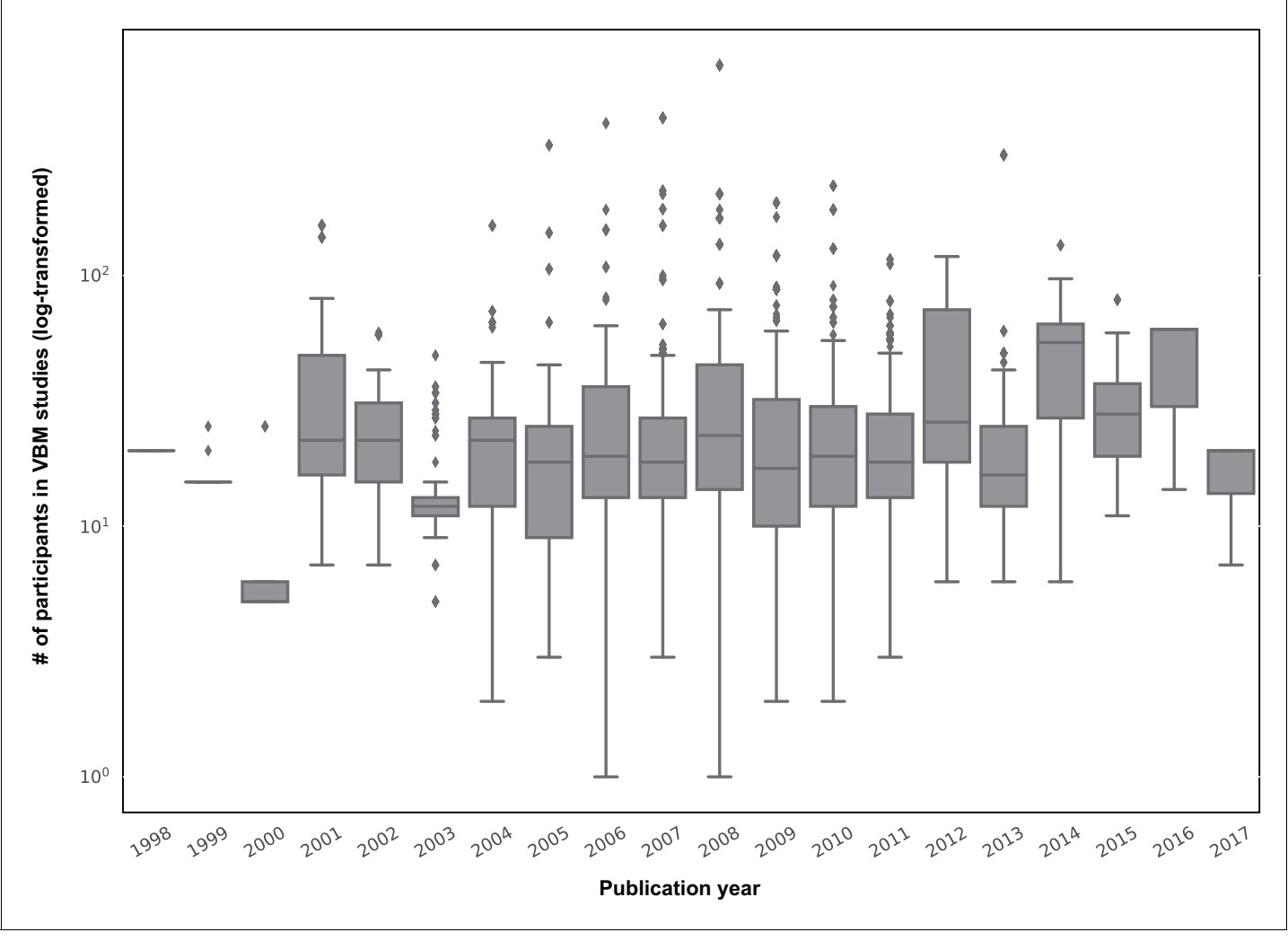

**Figure 5.** box-plots showing distribution of sample sizes (log-scale) of VBM studies by their publication year (data from the BrainMap database; see *Vanasse et al., 2018*). Each box shows the quantiles (25% and 75%) of the distribution and the gray horizontal line within each box, depicts the median of the distribution.

DOI: https://doi.org/10.7554/eLife.43464.010

researches: the dataset level, the analyses level, as well as at the post-publication and replication level.

*At the dataset level*, our study pointed out the need for big data samples to identify robust associations between psychological variables and brain structure, with sample size of at least several hundreds of participants. It should be acknowledged that this conclusion is easier to achieve than to implement in research practice. Nevertheless, large scale cohort datasets from healthy adult populations, such as eNKI used in the current study, human connectome project (HCP) (*WU-Minn HCP Consortium et al., 2013*) and UK-biobank (*Miller et al., 2016*) are now openly available, hence facilitating endeavor in that direction.

*At the analysis level*, we recommend the combined use of multivariate analyses, for comprehensive assessment of the spatial extent of associations and, univariate analyses, to facilitate interpretability, when studying brain structural correlates of psychological measures. Furthermore, we emphasis on the importance of *independent* confirmatory replications to provide evidence about the robustness of the effects.

Finally, *at the post-analysis level*, we concluded from our observations that publication of null findings should be more encouraged. In addition to directly shaping a more objective picture of

SBB-associations, these null-reports could contribute to new quantitative approaches. In particular, meta-analyses of published literature (*Vanasse et al., 2018*) would clearly benefit from such unbiased reports of null findings.

Sharing raw data would undoubtedly improve the problem of low statistical power, but if not possible, sharing the unthresholded statistical maps (e.g. through platforms such as Neurovault [*Gorgolewski et al., 2015*]) could also be a significant scientific contribution. In addition to directly contribute to our understanding of brain-behavior relationship, such efforts would open up new possibilities for estimating the correct size and extent of effects by integrating unthresholded statistical maps in the estimation of the effects sizes throughout the brain. Thus, we could hope that sharing initiatives will also contribute indirectly to more valid and insightful SBB studies in the remote future and hence to a better allocation of resources.

## Materials and methods

### Participants

Healthy adults' data were selected from the enhanced NKI (eNKI) Rockland cohort (*Nooner et al., 2012*). Data collection received ethics approval through both the Nathan Klein Institute and Montclair State University. Written informed consent was obtained from all participants.

We focused only on participants for which good quality T1-weighted scans was available along with timewise-corresponding psychological data. Exclusion criteria consisted of alcohol or substance dependence or abuse (current or past), psychiatric illnesses (eg. Schizophrenia) and current depression (major or bipolar). Furthermore, we excluded participants with missing information on important confounders (age, gender, education) or bad quality of structural scans after pre-processing, resulting in a total sample of 466 healthy participants (age: 48 ± 19, 153 male).

Replicability of SBB-associations within the clinical sample was investigated using a subsample drawn from the Alzheimer's Disease Neuroimaging Initiative (ADNI) database, which was launched in 2003 as a public–private partnership and led by Principal Investigator Michael W Weiner. The primary goal of ADNI has been to test whether serial magnetic resonance imaging (MRI), positron emission tomography (PET), other biological markers, and clinical and neuropsychological assessment can be combined to measure the progression of mild cognitive impairment (MCI) and early Alzheimer's disease (AD). For up-to-date information, see www.adni-info.org.

We used the baseline measurements from 371 patients (age: 71 ± 7, 200 male; 47 with significant memory complaint, 177 early MCI, 85 late MCI and 62 AD patients (defined based on ADNI diagnostic criteria, see http://adni.loni.usc.edu/wp-content/themes/freshnews-dev-v2/documents/clinical/ADNI-2_Protocol.pdf), in whom anatomical brain scans had been acquired in a 3Tesla scanner (from 39 different sites).

### Phenotypical measurements

#### Non-psychological measurements

Age and body mass index (BMI) are highly reliably assessed and their association with brain morphology has been frequently examined in previous studies (*Fjell et al., 2014*; *Kharabian Masouleh et al., 2016*; *Salat et al., 2004*; *Willette and Kapogiannis, 2015*).

Accordingly, they served here as the initial benchmarks among which SBB framework was tested in healthy individuals. In order to avoid large clusters that simultaneously cover several cortical and subcortical regions, we focused on local peaks of associations by increasing the voxel-level t-threshold of the statistical maps. The modified voxel-level t-threshold was set to 8 and 3, for defining age- and BMI-associated clusters, respectively. These *arbitrary* thresholds were chosen such that the very large clusters would divide into smaller ones, while still retaining the general spatial pattern of the significant regions.

#### Psychological measurements

The psychological measurements consisted in standard psychometrics and neuropsychological tests. The testing included: the attention network task (ANT) probing attention sub-functions (*Fan et al., 2002*), the Delis-Kaplan testing battery assessing different aspects of executive functions (*Delis et al., 2001*) (including trail-making test, color-word interference task, verbal fluency, 20

questions, proverbs and word-context task), the Rey auditory verbal learning task (RAVLT) (*Schmidt, 1996*) assessing verbal memory performance, as well as the WASI-II intelligence test (*Wechsler, 1999*). Psychological phenotyping also included anxiety (state and trait) (*Spielberger et al., 1970*) and personality questionnaires (*McCrae and Costa, 2004*) in the eNKI cohort. For each test, we used several commonly derived sub-scores to assess the replicability of their structural associations. For each psychological measure, participants whose performance deviated more than three standard deviation (SD) from mean of the whole sample were considered as outliers and thus were excluded from further analysis (See *Supplementary file 1*).

The list-learning task is a common measure of verbal learning performance and has been implemented using the same standard tool (RAVLT) in both the eNKI and the ADNI cohort. Previous studies have shown that the immediate-recall score (sum of recalled items over the first five trials) could be reliably predicted from whole brain MRIs in AD patients (*Moradi et al., 2017*). Since this score is a standard measure commonly used in healthy and clinical dataset and its relations to brain structure in clinical data has been previously suggested, in the current work we performed SBB with this score in the ADNI cohort as a 'conceptual benchmark'.

## MRI acquisition and preprocessing

The imaging data of the eNKI cohort were all acquired using a single scanner (Siemens Magnetom TrioTim, 3.0 T). T1-weighted images were obtained using a MPRAGE sequence (TR = 1900 ms; TE = 2.52 ms; voxel size = 1 mm isotropic).

ADNI, on the other hand, is a multisite dataset. Here we selected a subset of this data, which has been acquired in a 3.0 T scanner (baseline measurements from ADNI2 and ADNI GO cohort) from 39 different sites; see http://adni.loni.usc.edu/methods/documents/ for more information.

Both datasets were preprocessed using the CAT12 toolbox (*Gaser and Dahnke, 2016*). Briefly, each participant's T1-weighted scan was corrected for bias-field inhomogeneities, then segmented into gray matter (GM), white matter (WM), and cerebrospinal fluid (CSF) (*Ashburner and Friston, 2005*). The segmentation process was further extended for accounting for partial volume effects (*Tohka et al., 2004*) by applying adaptive maximum a posteriori estimations (*Rajapakse et al., 1997*). The gray matter segments were then spatially normalized into standard (MNI) space using Dartel algorithm (*Ashburner, 2007*) and further modulated. The modulation was performed by scaling the normalized gray matter segments for the non-linear transformations (only) applied at the normalization step. While this procedure ignores the volume changes due to affine transformation, it allows preserving information about individual differences in *local* gray matter volume. In other words, it re-introduces individual differences in local gray matter volume removed in the process of inter-subject registration and normalization. Finally modulated gray matter images were smoothed with an isotropic gaussian kernel of 8 mm (full-width-half-maximum).

## Statistical analysis

SBB-associations are commonly derived in an exploratory setting using a mass-univariate approach, in which a linear model is used to fit interindividual variability in the psychological score to GMV at each voxel. Inference is then usually made at cluster level, in which groups of adjacent voxels that support the link between GMV and the tested score are clustered together.

Replicability of thus-defined associations could be assessed by conducting a similar whole-brain voxel-wise exploratory analysis in another sample of individuals and comparing the spatial location of the significant findings that survive multiple comparison correction, between the two samples. Alternatively, replicability could be assessed, using a confirmatory approach, in which only regions showing significant SBB-association in the initial exploratory analysis, that is regions of interest (ROIs), are considered for testing the existence of the association between brain structure and the same psychological score in an independent sample. The latter procedure commonly focuses on a summary measure of GMV within each ROI and tests for existence of the SBB-association in the direction suggested by the initial exploratory analysis. Thus this approach circumvents the need for multiple comparison correction and therefore increases the power of replication.

Here we assessed replicability of associations between each behavioral measure and gray mater structure, using both approaches: the whole brain replication approach and the ROI replication approach, which are explained in details in the following sections.

## Replicability of whole brain exploratory SBB-associations

Whole-brain GLM analyses: 00 random subsamples (of same size) were drawn from the main cohort (eNKI or ADNI). Hereafter, each of these subsamples is called a 'discovery sample'. In each of these samples, SBB-associations were identified using the voxel-wise exploratory approach after controlling for confounders. This was done by using the general linear model (GLM) as implemented in the 'randomise' tool (https://fsl.fmrib.ox.ac.uk/fsl/fslwiki/Randomise), with 1000 permutations. Age, sex and education were modeled as confounders in the eNKI data. As the ADNI dataset is a multi-site study, we further added site and disease category as dummy-coded confounders to GLMs for the analyses in that dataset. Inference was then made using threshold-free cluster enhancement (TFCE) (*Smith and Nichols, 2009*), which unlike other cluster-based thresholding approaches does not require an arbitrary a-priori cluster forming threshold. Significance was set at p<0.05 (extent threshold of 100 voxels).

Spatial consistency maps and density plots: To quantify the spatial overlap of significant SBB associations over 100 subsamples, spatial consistency maps were generated. To do so, the binarized maps of all clusters that showed significant association in the same direction between each psychological score and GMV were generated (i.e. voxels belonging to a significant cluster get the value '1' and all other voxels were labeled '0') and added over all 100 subsamples. These aggregate maps denote the frequency of finding a *significant* association between the behavioral score and GMV, at each voxel. Accordingly, a voxel with value of 10 in the aggregate map has been found to be significantly associated with the phenotypical score in 10 out of 100 subsamples. Density plots were also generated to represent the distribution of values within each such map, that is the distribution of 'frequency of significant finding'. Hence, the spatial voxel-wise 'significance overlap maps' as well as density plots of the distribution of values within each map give indications of the replicability of 'whole brain exploratory SBB-associations' for each psychological score.

## Replicability of SBB-associations using confirmatory ROI-based approach

ROI-based confirmatory analyses: The replicability of the SBB associations was also evaluated with the ROI-based confirmatory approach. For each of the 100 discovery subsamples, an age- and sex-matched 'test sample' was generated from the remaining participants of the main cohort. In the clinical cohort the discovery and test pairs were additionally matched for 'site'. In this analysis, for each psychological variable, the significant clusters from the above-mentioned exploratory approach from every 'discovery sample' were used as a-priori ROIs. Average GMV over all voxels within the ROI was then calculated for each participant in the respective 'discovery 'and 'test' pair subsamples. Within each subsample, association between the average GMV and the psychological variable was assessed using ranked-partial correlation, controlling for confounding factors. The correlation coefficient was then compared between each discovery and test pair, providing means to assess 'ROI-based SBB replicability' rates for each psychological score. Accordingly, each ROI was examined only once, to identify if associations between average GMV in this ROI and the psychological score from the discovery subsample could be confirmed in the paired test sample. Replicability rates were quantified according to different indexes (see below) over all ROIs from 100 discovery samples, yielding a percentage of 'successfully replicated' ROIs based on each index.

Indexes of replicability:

Sign: First, we used a lenient definition of replication, in which we compared only the sign of correlation coefficients of associations within each ROI between the discovery and the matched-test sample. Accordingly, any effect that was in the same direction in both samples (even if very close to zero) was defined as a 'successful' replication.

Statistical significance: Another straightforward method for evaluating replication simply defines statistically significant effects (e.g. p-value<0.05) that are in the same direction as the original effects (from the discovery sample) as 'successful' replication. This criteria is consistent with what is commonly used in the psychological sciences to decide whether a replication attempt 'worked' (*Open Science Collaboration, 2015*). Yet, a key weakness of this approach is that it treats the threshold (p<0.05) as a bright-line criterion between replication success and failure. Furthermore, it does not quantify the decisiveness of the evidence that the data provides for and against the presence of the correlation (*Boekel et al., 2015*; *Wagenmakers et al., 2015*). However, such an estimation can be provided by using the 'Bayes factors'.

Bayes factor: To compare the evidence that the 'test subsample' provided for or against the presence of an association (H1 and H0, respectively), we additionally quantified SBB-replication within each ROI, using Bayes factors (*Jeffreys, 1961*). Similar to *Boekel et al. (2015)*, here we used the adjusted (one-sided) Jeffry's test (*Jeffreys, 1961*) based on a uniform prior distribution for the correlation coefficient. As we intended to confirm the SBB-associations defined in the discovery subsamples, the alternative hypothesis (H1) in this study was considered one-sided (in line with *Boekel et al., 2015*). We used implementation of the Bayes Factors for correlations from the R function available at http://www.josineverhagen.com/?page_id=76.

To facilitate the interpretation, Bayes factors (BF) were summarized into four categories as illustrated in the bar legend of *Figure 2*. A $BF_{01}$ lower than 1/3 shows that the data is three times or more likely to have happened under H1 than H0. Accordingly, this value defines the 'successful' replication.

## Investigation on factors influencing replicability of SBB-associations among healthy individuals

Sample size: In order to study the influence of sample size on the replicability of SBB-associations, for each psychological measure, the healthy sample (eNKI) was divided into discovery and test pairs at three different ratios: 70% discovery and 30% test, 50% discovery and 50% test and finally 30% discovery and 70% test. As mentioned earlier, in each case, the discovery and test counterparts were randomly generated 100 times in order to quantify the replication rates. For example, to assess the replicability of brain structural associations of age, in the case of '70% discovery and 30% test', the entire NKI sample (n = 466) was divided into a discovery group of n = 326 participants and an age- and sex-matched test pair sample of n = 138 and this split procedure was repeated 100 times. Similarly, for generating equal-sized discovery and test subsamples, 100 randomly generated age and sex matched split-half samples were generated from the main NKI cohort.

Due to the multi-site structure of the ADNI cohort, when generating unequal sized discovery and test samples, we did not achieve a good simultaneous matching of age, sex and site, while trying to maintain samples sizes in each subgroup reasonably large. Thus, in this cohort, we did not directly study the influence of the sample size and the replicability rates were only quantified for equal sized discovery and test samples (187 participants matched for age, sex and site between discovery and test pairs).

Effect size: Furthermore, to study the influence of the effect size on the replication rates, we focused on the effect sizes within each a-priori ROI in the discovery samples. Here we tested the following two assumptions:

1. ROIs with larger effect sizes in the discovery sample result in larger effect sizes in the test sample pairs (i.e. positive association between effect size in the discovery and test samples).
2. ROIs with larger effect sizes in the discovery sample are more likely to result in a 'significant' replication in the independent sample.

To test the first assumption, in the 'ROI-based SBB-replicability' the association between effect size in the discovery and test pairs were calculated for each psychological measure. These associations were calculated separately for the replicated (defined using 'sign' criterion) and not-replicated ROIs. We expected to find a positive association between discovery and confirmatory effect sizes, for the 'successfully replicated effects'.

To test the second assumption, for each ROI, we calculated its replication statistical power and compared it between replicated and not-replicated ROIs (here replication was defined using 'Statistical Significance' criterion). The statistical power of a test is the probability that it will correctly reject the null hypothesis when the null is false. In a bias-free case, the power of the replication is a function of the replication sample size, real size of the effect and the nominal type I error rate (α). In this study, the replication power was estimated based on the size of the effects as they were defined in the discovery sample and a significant threshold of 0.05 (one-sided) and was calculated using 'pwr' library in R (https://www.r-project.org).

These analyses were performed for each discovery-test split size, separately (i.e. 70–30%, 50–50% and 30–70% discovery-test sample sizes, respectively).

## Acknowledgements

Some of the data used in preparation of this article were obtained from the Alzheimer's Disease Neuroimaging Initiative (ADNI) database (adni.loni.usc.edu). As such, the investigators within the ADNI contributed to the design and implementation of ADNI and/or provided data but did not participate in analysis or writing of this report. A complete listing of ADNI investigators can be found at: http://adni.loni.usc.edu/wp-content/uploads/how_to_apply/ADNI_Acknowledgement_List.pdf.

This work was supported by the Deutsche Forschungsgemeinschaft (DFG, GE 2835/1–1, EI 816/4–1), the Helmholtz Portfolio Theme 'Supercomputing and Modelling for the Human Brain' and the European Union's Horizon 2020 Research and Innovation Programme under Grant Agreement No. 720270 (HBP SGA1) and Grant Agreement No. 785907 (HBP SGA2).

Clinical data collection and sharing for this project was funded by the Alzheimer's Disease Neuroimaging Initiative (ADNI) (National Institutes of Health Grant U01 AG024904) and DOD ADNI (Department of Defense award number W81XWH-12-2-0012). ADNI is funded by the National Institute on Aging, the National Institute of Biomedical Imaging and Bioengineering, and through generous contributions from the following: AbbVie, Alzheimer's Association; Alzheimer's Drug Discovery Foundation; Araclon Biotech; BioClinica, Inc; Biogen; Bristol-Myers Squibb Company; CereSpir, Inc; Cogstate; Eisai Inc; Elan Pharmaceuticals, Inc; Eli Lilly and Company; EuroImmun; F Hoffmann-La Roche Ltd and its affiliated company Genentech, Inc; Fujirebio; GE Healthcare; IXICO Ltd.; Janssen Alzheimer Immunotherapy Research and Development, LLC; Johnson and Johnson Pharmaceutical Research and Development LLC; Lumosity; Lundbeck; Merck and Co, Inc; Meso Scale Diagnostics, LLC; NeuroRx Research; Neurotrack Technologies; Novartis Pharmaceuticals Corporation; Pfizer Inc; Piramal Imaging; Servier; Takeda Pharmaceutical Company; and Transition Therapeutics. The Canadian Institutes of Health Research is providing funds to support ADNI clinical sites in Canada. Private sector contributions are facilitated by the Foundation for the National Institutes of Health (www.fnih.org). The grantee organization is the Northern California Institute for Research and Education, and the study is coordinated by the Alzheimer's Therapeutic Research Institute at the University of Southern California. ADNI data are disseminated by the Laboratory for Neuro Imaging at the University of Southern California.

## Additional information

### Funding

| Funder | Grant reference number | Author |
|---|---|---|
| Deutsche Forschungsgemeinschaft | GE 2835/1–1 | Shahrzad Kharabian Masouleh |
| Helmholtz-Gemeinschaft | Helmholtz Portfolio Theme 'Supercomputing and Modelling for the Human Brain' | Shahrzad Kharabian Masouleh Simon B Eickhoff Felix Hoffstaedter Sarah Genon |
| Horizon 2020 Framework Programme | 720270 (HBP SGA1) | Shahrzad Kharabian Masouleh Simon B Eickhoff Felix Hoffstaedter Sarah Genon |
| Horizon 2020 Framework Programme | 785907 (HBP SGA2) | Shahrzad Kharabian Masouleh Simon B Eickhoff Felix Hoffstaedter Sarah Genon |
| Deutsche Forschungsgemeinschaft | EI 816/4–1 | Simon B Eickhoff |

The funders had no role in study design, data collection and interpretation, or the decision to submit the work for publication.

### Author contributions

Shahrzad Kharabian Masouleh, Conceptualization, Data curation, Software, Formal analysis, Validation, Investigation, Visualization, Methodology, Writing—original draft, Project administration,

Writing—review and editing; Simon B Eickhoff, Conceptualization, Resources, Supervision, Funding acquisition, Project administration, Writing—review and editing; Felix Hoffstaedter, Resources, Data curation, Formal analysis, Project administration, Writing—review and editing; Sarah Genon, Conceptualization, Supervision, Funding acquisition, Methodology, Writing—original draft, Project administration, Writing—review and editing

### Author ORCIDs
Shahrzad Kharabian Masouleh (iD) http://orcid.org/0000-0003-4810-9542
Simon B Eickhoff (iD) https://orcid.org/0000-0001-6363-2759

### Ethics
Human subjects: Institutional Review Board Approval (IRBA) was obtained for this project at the Nathan Kline Institute (Phase I #226781 and Phase II #239708) and at Montclair State University (Phase I #000983A and Phase II #000983B). Written informed consent was obtained for all study participants. For ADNI data, IRBA was also obtained within each participating institute as well as informed consent forms were signed by each participant (see https://adni.loni.usc.edu/wp-content/uploads/2017/09/ADNID_Approved_Protocol_11.19.14.pdf for detailed information about ethical procedures for ADNI). Additionally, analysis of the data from both eNKI (Study No. 4039) and ADNI (Registration No. 2018114856) received ethical approval from the ethics committee of medical faculty at the University of Düsseldorf.

### Decision letter and Author response
Decision letter https://doi.org/10.7554/eLife.43464.015
Author response https://doi.org/10.7554/eLife.43464.016

## Additional files
### Supplementary files
• Supplementary file 1. Participants characteristics. Distribution of the raw phenotypical and psychological scores in the whole sample.
DOI: https://doi.org/10.7554/eLife.43464.011

• Supplementary file 2. Summary of the exploratory findings. For each discovery sample size, the number of clusters in which gray matter volume is positively or negatively associated with the tested psychological score is reported. Number of splits (out of 100) in which the clusters were detected are noted in parentheses.
DOI: https://doi.org/10.7554/eLife.43464.012

• Transparent reporting form
DOI: https://doi.org/10.7554/eLife.43464.013

### Data availability
All data used in this study, are openly available (eNKI, ADNI). The eNKI Rockland cohort data were downloaded from http://fcon_1000.projects.nitrc.org/indi/enhanced/. Users are first required to complete a Data Usage Agreement document before access to these data is granted (further details here http://fcon_1000.projects.nitrc.org/indi/enhanced/phenotypicdata.html). The ADNI data were downloaded from http://adni.loni.usc.edu/about/. Users must first request access before they can log in to the data archive and access is contingent on adherence to the ADNI Data Use Agreement (further details here http://adni.loni.usc.edu/data-samples/access-data/).

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
