## [Decision Letter]

Thank you for submitting your article "Empirical examination of the replicability of associations between brain structure and psychological variables" for consideration by *eLife*. Your article has been reviewed by two peer reviewers, and the evaluation has been overseen by a Reviewing Editor and Timothy Behrens as the Senior Editor. The reviewers have opted to remain anonymous.

The reviewers have discussed the reviews with one another and the Reviewing Editor has drafted this decision to help you prepare a revised submission.

Summary:

The work considers the reliability of behavioural correlations with brain structure, within a mass univariate linear modelling approach using Voxel Based Morphometry data. Using 446 healthy subjects and 371 patients, repeated subsamples are drawn and analysed to create maps of consistency of detected clusters in a whole-brain approach, and then a ROI approach is also used on matched "discovery" and matched "test" samples. 30 behavioural variables were evaluated, in addition to age and BMI used as high-power benchmark measures. The reproducibility of the brain-behavioural correlations was found to be very poor, with little overlap between discovery and test clusters. Reproducibility based solely on the sign of the correlation and (to a lesser extent) Bayes Factors showed better consistency, suggesting the presence of true effect but with very low detection power.

Although some other work on this topic has been carried out, the current study takes a rigorous approach, by assessing various aspects of replicability of SBB studies in different ways. Highlights of the study include:

- the use of two large, independent, samples (one with clinical subjects)

- assessing both the replicability of whole brain exploratory SBB associations, and that of an ROI-based confirmatory approach

- the use of two reference phenotypical measures (age and BMI) that serve as benchmark for replicability

- investigation of the effect of (discovery/test) sample size on replicability

Essential revisions:

There needs to be greater motivation for the particular analytic strategy taken in the paper. While the Introduction nicely covers the history of replicability in imaging and SSBs, and then describes all the analysis that is going to be done, there is a need for more motivation: Why all these analysis are done. What is the benefit in doing 30-70, 50-50 and 70-30 setups? Analysing spatial overlap? Why should we have direction testing, sign testing, and Bayes' testing? What are the scientific questions that will be answered by these questions? How has previous literature lead us to answer these questions. These concerns can be highlighted with a bit of reorganizing the Introduction. Ideally, each analysis decision in the Results should be set up by a few sentences in the Introduction as to why it is necessary to see the results of such step. While carefully described, the motivation of the very lenient replicability measure "by sign" should be amplified (at first glance, one wonders what could be the added value is since one may expect an enormous amount of correlation coefficients of about zero, carrying hardly any information regarding replicability).

The Discussion is missing a well-identified limitations section. For example, while sample sizes of the current study are relatively large (and maybe large as compared to most published works in this area), the conclusions about poor replicability are limited to these sample sizes; e.g. if you had used the Human Connectome Project or larger samples, you would have potentially discovered the N where reliability 'kicks in' for more of the behavioral measures. Another limitation might be the fact that only one implementation is used: cluster-based analysis, whereas not all studies employ this approach. Moreover, the results (and conclusions) are also limited with respect to parameter settings, such as smoothing kernel width and the use of modulated gray matter segments (instead of, e.g. surface based analyses).

The role of multivariate statistical methods, which are growing in use, was mentioned only in passing in the Discussion. Please make a note already in the Introduction about multivariate side-by-side with the mass-univariate approach and how they have different replicability properties, emphasising that this work focusses on the mass-univariate approach. In the Discussion, the recommendation comes off as a gap in the paper. While the reviewers agree that enough analysis has been done, the phrasing of the multivariate discussion (subsection “Poor spatial overlap of SBB across resampling: possible causes and recommendations”, last paragraph) could be reorganized so that the reader is not left with the question "Why weren't these extra interesting ideas tried out?" Perhaps suggestions can be given as to how those hypotheses/recommendations could be tested in the future.

In the Discussion a clear discussion on sample size and power vis-à-vis correlation is needed. For example, at first blush the sample sizes in this study and in the discipline are simply too small. For example, note that in the UK Biobank the strongest correlation between cognition and T1 was found to be r=0.10 https://www.nature.com/articles/nn.4393/figures/6, Figure 6B), indicating a need for about N=800 subjects to attain 80% power at 0.05 uncorrected. Does your analysis indicate a minimum r that is needed for the sample sizes considered?

---

## [Author Response]

Essential revisions:There needs to be greater motivation for the particular analytic strategy taken in the paper. While the Introduction nicely covers the history of replicability in imaging and SSBs, and then describes all the analysis that is going to be done, there is a need for more motivation: Why all these analysis are done. What is the benefit in doing 30-70, 50-50 and 70-30 setups? Analysing spatial overlap? Why should we have direction testing, sign testing, and Bayes' testing? What are the scientific questions that will be answered by these questions? How has previous literature lead us to answer these questions. These concerns can be highlighted with a bit of reorganizing the Introduction. Ideally, each analysis decision in the Results should be set up by a few sentences in the Introduction as to why it is necessary to see the results of such step. While carefully described, the motivation of the very lenient replicability measure "by sign" should be amplified (at first glance, one wonders what could be the added value is since one may expect an enormous amount of correlation coefficients of about zero, carrying hardly any information regarding replicability).

We agree with the reviewers and thus have now extended our Introduction and elaborated more on the motivations behind each step. The modified parts of the “Introduction” now read as:

*“*In another study we demonstrated lack of robustness of the pattern of correlations between cognitive performance and measures of gray matter volume (GMV) in a-priori defined sub-regions of the dorsal premotor cortex in two samples of healthy adults (Genon et al., 2017). […] For this purpose, a subsample of patients drawn from Alzheimer's Disease Neuroimaging Initiative (ADNI) database were selected, in which replicability of structural associations of immediate-recall score from Rey auditory verbal learning task (RAVLT) (Schmidt, 1996) was assessed (see Materials and methods). Due to availability of the same score within the healthy cohort, this later analysis is used as a ‘conceptual” benchmark.’”

The Discussion is missing a well-identified limitations section. For example, while sample sizes of the current study are relatively large (and maybe large as compared to most published works in this area), the conclusions about poor replicability are limited to these sample sizes; e.g. if you had used the Human Connectome Project or larger samples, you would have potentially discovered the N where reliability 'kicks in' for more of the behavioral measures. Another limitation might be the fact that only one implementation is used: cluster-based analysis, whereas not all studies employ this approach. Moreover, the results (and conclusions) are also limited with respect to parameter settings, such as smoothing kernel width and the use of modulated gray matter segments (instead of, e.g. surface based analyses).

We thank the reviewers for pointing out this issue. We have now added a new section with the subheading “Limitations”, in which we discussed the most relevant limitations of our study and their possible impact on the conclusions drawn.

However, it is worth noting that when considering using a dataset such as the Human connectome project, with related individuals, one would need to carefully limit the sample to unrelated individuals, only, to avoid biasing the replication attempt and to provide a generalizable estimation of the replication of SBB-associations. As HCP consists of ~460 families, it would not be of huge advantage regarding the sample size, compared to the eNKI cohort that is used in the current manuscript. However we agree with the spirit of the comment raised by the reviewers that a larger sample such as the UK Biobank, which is a very valuable and unique resource, could be of great advantage to study influence of sample size on the replicability of structure-brain-behavior associations. Accordingly, we emphasize on this point both within the newly added “limitation section” as well as the “summary and conclusions” of our revised manuscript.

The limitation section now reads as:

“Limitations:

When interpreting our results, it should be noted that, in order to keep large sample sizes for the exploratory replication analyses, the discovery subsamples were not necessarily designed to be independent from each other. […] Future studies should therefore investigate to which extend our replicability rates are reproduced with different data preprocessing pipelines and analyses approaches.”

The role of multivariate statistical methods, which are growing in use, was mentioned only in passing in the Discussion. Please make a note already in the Introduction about multivariate side-by-side with the mass-univariate approach and how they have different replicability properties, emphasising that this work focusses on the mass-univariate approach. In the Discussion, the recommendation comes off as a gap in the paper. While the reviewers agree that enough analysis has been done, the phrasing of the multivariate discussion (subsection “Poor spatial overlap of SBB across resampling: possible causes and recommendations”, last paragraph) could be reorganized so that the reader is not left with the question "Why weren't these extra interesting ideas tried out?" Perhaps suggestions can be given as to how those hypotheses/recommendations could be tested in the future.

We agree with the reviewers on the valuable role that application of multivariate approaches can play in identification of the relationships between brain and behavior. Specifically in the Discussion we explicitly discuss the evidence from the literature supporting multivariate nature of the relationship between brain and behavior. Nevertheless, as now stated in our Introduction, the mass-univariate approach was historically dominant and remains as the main workhorse of the neuroimaging literature. Univariate methods are widely implemented in common neuroimaging tools. Easy interpretation of their outcomes and their historical precedence have made them as the first choices for many researchers, specifically when focusing on specific behavior features in small and moderate samples. Acknowledging the popularity of the univariate approach, in the current study, we aim to empirically show the replication rate of findings from *such* studies. Furthermore, to provide a balance consideration of mapping methods and a constructive discussion, we suggested certain recommendations at multiple levels of a project to improve the replicability of the claimed effects. In particular, at the analysis level, we bring concrete examples of studies that combine the multivariate and univariate approaches.

The modified sections, now read as:

Introduction:

*“*While in the recent years multivariate methods are frequently recommended to explore the relationship between brain and behavior (Cremers et al., 2017; Smith and Nichols, 2018), SBB-association studies using these approaches remain in minority. […] The current study, therefore, focused on the assessment of replicability of SBB-associations using the latter approach.*”*

Discussion:

*“*It is worth noting that similar complexity and uncertainty have been described for task-based functional associations studies (Cremers et al., 2017; Turner et al., 2018). […] Although it does not provide any protection against the influence of noise that may affect both approaches, this solution may help to reduce the false negatives.*”*

In the Discussion a clear discussion on sample size and power vis-à-vis correlation is needed. For example, at first blush the sample sizes in this study and in the discipline are simply too small. For example, note that in the UK Biobank the strongest correlation between cognition and T1 was found to be r=0.10 https://www.nature.com/articles/nn.4393/figures/6, Figure 6B), indicating a need for about N=800 subjects to attain 80% power at 0.05 uncorrected. Does your analysis indicate a minimum r that is needed for the sample sizes considered?

As the reviewers aptly mention, the sample sizes in the discipline are generally very small (as we also show in Figure 5 of our manuscript.). So an important aspect of our study was to show the expected replicability rate of SBB-findings in such sample sizes. In light of multiple analyses results, we also emphasize in the Discussion of our manuscript that the sample sizes of 200-300 participants are still far too small to reliably characterize replicable structure-brain-behavior associations among healthy individuals. This is an important factor resulting in exaggerated effect size (correlation coefficient) in the exploratory analyses. For example, considering the results of perceptual reasoning task, for which the whole-brain exploratory analyses showed fairly replicable associations with GMV, confirmatory ROI analyses (even with sample size = 326) showed replicable association (defined as significant association with the same “sign”) in less than 30% of ROIs. Considering this value as the realistic power of replication, with a sample of 326 participants, this would suggest a maximum correlation coefficient of 0.08, within the discovery cohort. Figure 3 clearly shows correlation coefficients of all above 0.1 and even as large as 0.4 (x-axis), pointing towards exaggerated effect size estimation in the exploratory cohorts. Therefore, considering the objective of our study (i.e. characterizing the replication rate of association between brain structure and normal variation in behavior among healthy individuals) and such exaggerated effects from the exploratory analyses, we believe that defining a minimum r-value (as questioned by the reviewer), without contemplating on improving different aspects of analysis (as discussed in our manuscript) would not be informative.

As suggested by the reviewers, these observations are calling for the use of much larger samples, such as the suggested UK Biobank cohort, in order to reliably identify replicable SBB-effects. We now emphasis this factor more strongly in our manuscript, by showing its implication both in our results and comparing our sample sizes to the commonly used samples in the literature. Yet, we also emphasize on the importance of other factors, namely the object of the study itself and the reliability of the measures under study as noteworthy factors.

The modified section on sample size and power in relation to our results, now reads as:

“Higher power is defined as increased probability of finding effects that are genuinely true. […] Yet, while many replication studies straightforwardly blame the sample size of the original studies, it is important to keep in mind that a replication failure might also come from a too small sample size of the replication study (Muhlert and Ridgway, 2016).”

And:

“When interpreting our results, it should be noted that, in order to keep large sample sizes for the exploratory replication analyses, the discovery subsamples were not necessarily designed to be independent from each other.. […] Recent advancements through data collection from much larger number of participants, such as UK Biobank (Miller et al., 2016) are promising opportunities for overcoming these limitations in future replication studies.”